# Genome-Wide Identification and Expression Analysis of WRKY Transcription Factors in *Siraitia siamensis*

**DOI:** 10.3390/plants12020288

**Published:** 2023-01-07

**Authors:** Detian Mu, Wenqiang Chen, Yingying Shao, Iain W. Wilson, Huan Zhao, Zuliang Luo, Xiaodong Lin, Jialong He, Yuan Zhang, Changming Mo, Deyou Qiu, Qi Tang

**Affiliations:** 1College of Horticulture, Hunan Agricultural University, Changsha 410128, China; 2CSIRO Agriculture and Food, Canberra, ACT 2601, Australia; 3School of Traditional Chinese Medicine, Capital Medical University, Beijing 100069, China; 4Institute of Medicinal Plant Development, Chinese Academy of Medical Sciences, Peking Union Medical College, Beijing 100193, China; 5Guangxi Crop Genetic Improvement and Biotechnology Laboaratory, Guangxi Academy of Agricultural Sciences, Nanning 530007, China; 6State Key Laboratory of Tree Genetics and Breeding, Research Institute of Forestry, Chinese Academy of Forestry, Beijing 100091, China

**Keywords:** *Siraitia siamensis*, WRKY transcription factor, bioinformatics analysis, expression pattern, cold stress

## Abstract

WRKY transcription factors, as the largest gene family in higher plants, play an important role in various biological processes including growth and development, regulation of secondary metabolites, and stress response. In this study, we performed genome-wide identification and analysis of WRKY transcription factors in *S. siamensis*. A total of 59 *SsWRKY* genes were identified that were distributed on all 14 chromosomes, and these were classified into three major groups based on phylogenetic relationships. Each of these groups had similar conserved motifs and gene structures. We compared all the *S. siamensis SsWRKY* genes with *WRKY* genes identified from three diverse plant species, and the results implied that segmental duplication and tandem duplication play an important roles in the evolution processes of the *WRKY* gene family. Promoter region analysis revealed that *SsWRKY* genes included many *cis*-acting elements related to plant growth and development, phytohormone response, and both abiotic and biotic stress. Expression profiles originating from the transcriptome database showed expression patterns of these *SsWRKY* genes in four different tissues and revealed that most genes are expressed in plant roots. Fifteen *SsWRKY* genes with low-temperature response motifs were surveyed for their gene expression under cold stress, showing that most genes displayed continuous up-regulation during cold treatment. Our study provides a foundation for further study on the function and regulatory mechanism of the *SsWRKY* gene family.

## 1. Introduction

WRKY transcription factors are one of the largest families of transcriptional regulators in plants and are integral to signaling webs that modulate many important plant processes [1]. WRKY refers to a highly conserved protein domain, the WRKY domain, with nearly 60 amino acids [2]. The most significant feature of these proteins is that they all contain a WRKYGQK sequence in N-terminal and a C_2_H_2_ or C_2_HC (C-X_4-5_-C-X_22-23_-H-X_1_-H or C-X_7_-C-X_23_-H-X_1_-H) zinc finger motif at the C-terminal [3,4]. Previous studies have shown that several variants of the WRKYGQK motif exist, including WRKYGEK and WRKYGKK. A typical WRKY motif has therefore been suggested, embracing the consensus sequence W(R/K)(K/R)Y [5]. According to the number of WRKY domains and the type of zinc-finger motif, WRKY proteins can be divided into three major groups, Group I, Group II, or Group III [6]. Those in Group I contain two WRKY domains and one C_2_H_2_ (C-X_4-5_-C-X_22-23_-H-X_1_-H) zinc finger motif; Group II includes proteins with one WRKY domain and a C_2_H_2_ zinc finger motif, and can be further divided in five subgroups (IIa, IIb, IIc, IId and IIe) according to the primary amino acid sequence [7]. Group III proteins have one WRKY domain and one C_2_HC zinc finger motif. WRKY proteins are capable of binding to the W-box cis-elements [(T)TGAC(C/T)] in the promoter region of the target genes, and can activate, repress or depress transcription, alone or in combination with other regulators [8,9,10]. Owing to their structure, WRKYs have many important biological functions.

Many studies have shown that WRKY proteins play critical roles in plant growth and development, biotic and abiotic stress response, senescence, secondary metabolism, and phytohormone signaling [11,12,13,14,15]. For example, three *Arabidopsis thaliana* Group III WRKY transcription factors, *AtWRKY46*, *AtWRKY54*, *AtWRKY70* are positively involved in brassinosteroid-regulated plant growth [16]. In *Solanum lycopersicum*, *SlWRKY8* promotes resistance to pathogen infection, and mediates tolerance of drought and salt stress [17]. In *Polygonatum odoratum*, expression of *PoWRKY1* is induced by cold and drought stress, and overexpression in *A. thaliana* improved seed germination vigor and root growth of transgenic plants under cold stress and drought [18]. At least 15 *WRKY* genes in *Acer truncatum* are expressed in cold stress [19]. Furthermore, WRKY40-D and WRKY42-B from *Taraxacum antungense* have been demonstrated to promote initiation of leaf senescence by promoting the jasmonic acid biosynthesis pathway [20,21]. Overexpression of *SmWRKY1* in *Salvia miltiorrhiza* and of *CrWRKY1* in *Catharanthus roseus* were each shown to positively promote biosynthesis of secondary metabolites, elevating transcripts from genes coding for enzymes in biosynthetic pathways and resulting in their increased accumulation [22,23]. In *Hylocereus polyrhizus*, *HPWRKY3* activates sucrose metabolic genes, thus increasing sugar accumulation in pitaya fruit [24]. Many WRKYs have been identified in various plants, including 72 in *A. thaliana* [25], 46 in *Ophiorrhiza pumila* [5], 61 in *Cucumis sativus* [26], and 63 in *Citrullus lanatus* [27]. However, to date there has been no information on WRKY transcription factors in *Siraitia siamensis*.

*S. siamensis* has been used in traditional Chinese medicine for many years as treatment for lung congestion, colds, and sore throats [28,29]. Its main bioactive ingredients are mogrosides, a kind of triterpenoid sweetener. Among these, mogroside V is a herbal sweetener used in the management of diabetes, and its content is higher in *S. siamensis* than in *Siraitia grosvenorii*. When Siamenoside I is separated from the fruit of *S. siamensis*, it is approximately 560 times sweeter than sucrose [30], ~1.4-fold sweeter than aspartame, and is used as a sugar substitute for diabetic patients [31]. *Siraitia* plants (*S. grosvenorii and S. siamensis*) are important perennial vines of the Cucurbitaceae family, with high economic value [32]. The natural distribution range of *S. siamensis* is mainly within Guangxi, Nanning province in China, Thailand, and Vietnam. Compared with *S. grosvenorii*, *S. siamensis* offers many advantages including disease resistance and improved fruit-setting percentage [33]. Therefore, *S. siamensis* may in future become an important source of mogroside V. In recent years, there have been many studies on *S. grosvenorii* [34,35,36], but research on the molecular biology of *S. siamensis* has to date been focused only on molecular markers [31]. Genome analysis of *S. siamensi* is lacking, especially for gene families that may play an important role in economically important aspects of this species.

In this study, based on the genome data of *S. siamensis* from NCBI (https://dataview.ncbi.nlm.nih.gov/object/SRR22947134?reviewer=qmfs7mc075pohiv011jqjceh94, accessed on 2 January 2023), we comprehensively identified and characterized the WRKY gene family with respect to chromosomal location, classification, collinearity, cis-acting elements, expression patterns, and potential protein interactions. Furthermore, 15 *SsWRKYs* under cold stress were analyzed by qRT-PCR. The results lay a foundation for deeper research on *WRKY* genes and provide a new perspective on the function of *WRKY* genes under abiotic and biotic stress.

## 2. Results

### 2.1. Identification and Characterization of SsWRKY in S. siamensis

To identify potential *WRKY* genes, the *S. siamensis* genome database (https://dataview.ncbi.nlm.nih.gov/object/SRR22947134?reviewer=qmfs7mc075pohiv011jqjceh94, accessed on 2 January 2023) was screened using HMM3.0, and a total of 61 candidates were identified. Of these, two candidate *SsWRKY* with incomplete WRKY domains were not considered to be *SsWRKY*. These were classified as *SsWRKY-like* genes, and were removed from our further analysis. The remaining 59 *SsWRKY* genes were named *SsWRKY1*–*SsWRKY59* according to their position on chromosome of *S. siamensis*. The detailed characterization of 59 *SsWRKY* genes is listed in Table 1. The protein length ranged from 128 (SsWRKY14) to 745 (SsWRKY39) amino acids, and the corresponding predicted molecular weights varied from 14.59 kDa to 81.91 kDa, with an average of 40.46 kDa. The theoretical pI ranged from 4.83 (SsWRKY51) to 9.92 (SsWRKY23). Subcellular localization prediction assigned all 59 proteins to the nucleus. The great majority of proteins were considered unstable, except SsWRKY3, SsWRKY10, SsWRKY13, SsWRKY22, SsWRKY23, and SsWRKY50 proteins.

### 2.2. Classification and Phylogenetic Relationships of SsWRKYs

Combining 70 AtWRKY protein sequences with 59 SsWRKY protein sequences, MEGA7.0 software was employed to construct the phylogenetic tree and classify the sequences into three groups based on the features of the WRKY gene family in *A. thaliana* (Figure 1). Eleven SsWRKY proteins were classified into Group I, forty SsWRKY proteins were classified into Group II and were further divided into five subgroups, with 4, 5, 16, 7, and 8 members in subgroups IIa–IIe, respectively. Group III consisted of the remaining eight SsWRKY proteins.

Multiple sequence alignments were used to identify the structural characteristics of SsWRKY proteins. As is shown in Figure 2, the majority of the 59 SsWRKY proteins (48/59, 81%) had a single conserved WRKY domain, and the remaining 11 proteins (19%) contained two WRKY domains. The highly conserved WRKY motif represented by 58 WRKYGQK and only one WRKYGKK (SsWRKY10), was identified in 59 SsWRKY proteins, similar to results of studies from other species including *Isatis indigotica* [37], *O. pumila* [5], and tomato [38]. All Group I members contained two complete WRKY domains, including a WRKYGQK sequence and a C_2_H_2_-like zinc-finger motif. All 40 Group II proteins contained one WRKY domain and a C_2_H_2_ motif (C-X_4-5_-C-X_22-23_-H-X_1_-H). Eight members of Group III had a WRKYGQK sequence and the C_2_HC zinc finger motif (C-X_7_-C-X_23_-H-X_1_-H).

### 2.3. Gene Structure and Conserved Motif Analysis

To better illustrate the similarity and diversity of motif composition among SsWRKYs, the MEME program was employed to analyze their conserved motifs. A total of ten conserved motifs were identified in the WRKY proteins of *S. siamensis*, ranging from 15 to 38 amino acids (Figure 3d), as shown in Figure 3 and Appendix A. Motifs 1 and 2 were extensively distributed in all SsWRKY proteins, corresponding to the conserved WRKY domain. Motifs 3, 5, and 10 were unique to Group I. Motif 6 was found mainly in Groups IIa and IIb; Motifs 7 and 8 were mainly distributed in Group IIe, Group IId, and Group III; Motif 9 was found only in Groups IIa and IIb (Figure 3b). In general, almost all the SsWRKYs that belonged to the same subgroups displayed similar motif composition, implying that these SsWRKY proteins have analogous functions.

To help clarify the evolution of *SsWRKY* family genes and their genetic structural diversity, the exon–intron organization of the coding sequences of *SsWRKY* genes was examined. As is shown in Figure 3c, the number of the exons in SsWRKYs varied from two to seven exons (7 with two exons, 29 with three exons, 8 with four exons, 10 with five exons, 4 with six exons), with SsWRKY21 containing the highest number of predicted exons (*n* = 7). These results show that the genetic structural diversity among *SsWRKY* genes may be the result of exon-loss and -gain events during the evolution of the *SsWRKY* gene family. *SsWRKY* genes from the same group showed similar exon/intron structures, as in Group IIe. In addition, *SsWRKY* genes with similar structures clustered together, as in Group III and Group IIe which contained three exons and two introns (or three introns) except for *SsWRKY24*, *SsWRKY11*, and *SsWRKY36*.

### 2.4. Chromosomal Location and Collinearity Analysis of SsWRKY Genes

As displayed in Figure 4, all 59 *SsWRKY* genes were mapped to 14 chromosomes of *S. siamensis*. Remarkably, although *WRKY* genes were not distributed evenly on every chromosome, they were present on each chromosome. There was no apparent relationship between the number of *SsWRKY* genes and chromosome length (Figure 4). *SsWRKY* genes were most abundant on Chr 3, with eight genes, and least abundant on Chr 12, 13, and 14, each with two genes. Collinear blocks within the *S. siamensis* genome were investigated to identify the relationship between *SsWRKY* genes and gene duplication events (Figure 5a). Twenty-four gene pairs were detected in the *S. siamensis* genome and were located on different chromosomes, suggesting that segmental duplications in these locations potentially contributed to expansion of the *SsWRKY* family. In addition, only one tandem duplication gene pair was identified, *SsWRKY8* and *SsWRKY9* located on Chr 2 (Figure 4). Some reports have indicated that tandem gene duplication is an important reason for gene clustering [39]. To enquire into the phylogenetic mechanisms of the *SsWRKY* gene family, three collinear maps of *S. siamensis* were constructed using three representative species, one monocotyledon (*O. sativa*) and two dicotyledons (*A. thaliana* and *C. sativus*) (Figure 5b–e). A total of 56 *SsWRKY* genes showed a collinear relationship with *C. sativus* (*n* = 56), followed in terms of frequency by *A. thaliana* (*n* = 50), and *O. sativa* (*n* = 20) (Appendix A). There were 85, 72, and 29 orthologous pairs between the other three species, respectively (*C. sativus*, *A. thaliana,* and *O. sativa*). Syntenic pairs were identified with three other species (with 13 *SsWRKY* genes, SsWRKY7, 8, 11, 12, 16, 26, 36, 38, 40, 45, 47, 49), implying that these gene pairs may have been present before ancestral divergence and played a key role in the evolution of the *SsWRKY* gene family.

### 2.5. Analysis of Cis-Acting Elements of SsWRKY Gene Family

To further our understanding of the potential function and regulation of WRKY transcription factors in *S. siamensis*, nearly 2.0 kb of sequence regions upstream of the translation initiation site was extracted to identify the *cis*-acting elements, using PlantCARE software. Almost all *SsWRKY* genes had at least one *cis*-acting element in the promoter region. These *cis*-acting elements were further classified into three categories, plant growth and development, phytohormone-responsive, and abiotic and biotic stress (Figure 6). The first category, plant growth and development, included light-responsive and development-related elements. Various light-responsive elements were presented in the promoter region of *SsWRKY* genes, containing Box4, G-box, GT1-motif, GATA-motif, Sp1, etc. The Box 4 element (203) was the most abundant *cis*-acting element found in *SsWRKY* promoters. There were also many elements related to development, including CAT-box, MSA-like, and O_2_-site. The second category contained phytohormone-responsive elements, composed of the TGACG-motif, TCA-element, ABRE, GARE-motif, etc. For example, the promoter region of *SsWRKY1* had ABRE, AuxRR-core, CGTCA-motif, GARE-motif, and TGACG-motif, indicating that these *cis*-acting elements are related with ABA, auxin, methyl jasmonate (MeJA), gibberellin, and salicylic acid responsiveness. It is speculated that *SsWRKY1* may therefore be involved in multiple hormone responses. The third category included *cis*-acting elements related to abiotic and biotic stress, including antioxidant response element (ARE), low temperature response (LTR) element, drought inducibility elements (MBS), wound responsive elements (Wun-motif), and TC-rich repeats. In *S. siamensis*, 139 ARE elements were present in 53 promoters (90%) of *SsWRKY* genes. In contrast, Wun-motif was the least abundant element in *S. siamensis*. LTR elements were found in 15 promoters of *SsWRKY*. Based on the strong induction in response to cold-stress treatment, *SsWRKY44* and *SsWRKY56* had four copies of the LTR element. Remarkably, the *SsWRKY37* gene did not contain any elements related to abiotic or biotic stress. In general, these results demonstrate that the *SsWRKY* genes might be associated with growth and development, hormones, and stress responses. The analysis of the putative *cis*-acting elements of the *SsWRKY* gene family might improve understanding of stress response, especially low temperature stress in *S. siamensis*.

### 2.6. Protein Interaction Network and Gene Co-Expression of SsWRKYs

String software was used to construct a protein–protein interaction network of 18 SsWRKY proteins based on the *A. thaliana* association model. In the interaction network diagram (Figure 7a and Appendix A), connected genes might have close functional relationships. The thicker the connecting line, the stronger is the predicted interaction between the two proteins. Among the 18 SsWRKY proteins, there were three proteins from Group I, eleven proteins belonging to Group II, and four proteins in Group III. In addition, SsWRKY53, SsWRKY54, SsWRKY11, SsWRKY12, SsWRKY38, SsWRKY47, SsWRKY3, and SsWRKY39 were shown to have stronger predicted interaction networks with other proteins. These proteins might play a key role in the whole process of WRKY regulation. The results of *SsWRKYs* co-expression showed that most genes were co-expressed, with the red color representing higher co-expression levels. As shown in Figure 7b, the cluster members *SsWRKY4*, *5*, *27*, *28*, *41*, *42*, *44*, *45*, *46*, *47* had high co-expression levels. Among them, the co-expression levels between *SsWRKY44-47* and *SsWRKY4-5*, *SsWRKY27-28*, *SsWRKY41-42* were significantly higher than those among other members.

### 2.7. Expression Profiles of the SsWRKY Genes in Different Tissues

The expression levels of the 59 *SsWRKY* genes were investigated in four different tissues of *S. siamensis* (roots, stems, tubers, and leaves), using the *S. siamensis* transcriptome database (https://dataview.ncbi.nlm.nih.gov/object/PRJNA917212?reviewer=qmfs7mc075pohiv011jqjceh94, accessed on 2 January 2023). All 59 *SsWRKY* genes were found to have significant expression in at least one tissue, and changes in expression were found between the different tissues surveyed, except for *SsWRKY12* (Figure 8). The results showed that *SsWRKY* genes had the highest expression levels in the roots; 37% (22/59), 31% (18/59), 29% (17/59), and 2% (1/59) of the *SsWRKY* genes displayed their highest expression levels in roots, tubers, stems, or leaves, respectively. These differences in the expression profiles of the *WRKY* genes indicate potentially different roles in these tissues.

### 2.8. Expression Analysis of SsWRKY Genes under Cold Stress

In order to examine the expression patterns of *SsWRKY* genes potentially associated with responses to low temperatures, 15 *SsWRKY* genes containing LTR *cis*-acting elements in their promoter were selected and surveyed for their expression levels during different stages of induced low-temperature stress (4 °C) (0, 6, 12, 24, 36 and 48 h). The expression of the selected *SsWRKY* genes could be classified into three groups according to their expression patterns, as shown in Figure 9. The highest expression levels in the majority of selected *SsWRKY* genes (*SsWRKY12*, *SsWRKY13*, *SsWRKY16*, *SsWRKY20*, *SsWRKY29*, *SsWRKY30*, *SsWRKY47*, *SsWRKY49*, *SsWRKY57*) were found after exposure to low temperature for 48 h. *SsWRKY44*, *SsWRKY51*, and *SsWRKY56* showed their highest levels at 6 h, and then gradually decreased, returning to basal expression levels after 48 h cold treatment, whereas *SsWRKY41*, *SsWRKY42*, *SsWRKY53* were highly expressed at 24 h, falling again to initial expression levels after 48 h cold treatment.

## 3. Discussion

*S. siamensis* is important for its economic and medicinal value [31]. However, little research has been conducted at the molecular level due to a lack of genome and transcriptome data. Studies in many plant species have shown that the *WRKY* gene family is the largest transcription factor family, and members often participate in important plant processes, especially stress response [40]. Based on whole genome sequences, the structure and function of the *WRKY* gene family has been widely investigated in several species including *A. thaliana* [41], *O. sativa* [42], *C. sativus* [26], and *Cucumis melon* [27]. However, there have been no equivalent studies in *S. siamensis*. Genome sequencing of *S. siamensis* has been recently completed (unpublished), providing a means for performing genome-wide analysis. In this study, we provide identification and analysis of the *SsWRKY* gene family for the first time.

Based on the number of the conserved WRKY domains and the feature of the zinc finger motif, the *WRKY* gene family was classified into three groups. Following phylogenetic analysis with *AtWRKY* genes, the 59 *SsWRKYs* were divided into three main groups, 11 SsWRKY proteins belonging to Group I, 40 SsWRKY proteins classified into Group II, accounting for the largest proportion (68%), indicating that this group may have experienced gene duplication events during its evolution, and the remaining eight SsWRKY proteins belonged to Group III. Similar results have been reported from *Zea mays* [43], *C. sativus* [26], *Coffea canephora* [44]. Furthermore, variation of the conserved WRKY domain (WRKYGKK) was observed in the SsWRKY10 protein sequence from the subgroup IIc. A similar situation has also been described in other plants such as *O. pumila* (OpWRKY16, OpWRKY27, OpWRKY28, OpWRKY30, and OpWRKY47), and *C. sativus* (CsWRKY10 and CsWRKY47) [5,26]. Previous research has shown that the WRKYGQK domain can bind to W-box *cis*-elements in the promoter region to interact with target genes [22], and changes of the WRKYGQK domain may influence their binding specificity. Therefore, the function and DNA-binding specificities of the SsWRKY10 protein should be further analyzed. Based on the analysis of conserved motifs, WRKY members in the same group or subgroup were found to have similar motif composition. Almost all SsWRKY contained motif 1 and motif 2, which may be a core element retained during evolution. The conserved motifs of IId and IIe subgroups were similar, which may suggest their genetic relationship through evolution.

Gene duplication events, including tandem, segmental, and whole-genome duplications, are important for acquiring new genes and enabling expansion on gene families in organisms [45]. Among 69 *SsWRKY* genes, only *SsWRKY8* and *SsWRKY9* displayed tandem duplication (Group IIa). Among the *SsWRKY* genes, the number of segmental duplications exceeded the number of tandem duplications. Twenty-four gene pairs were detected in the *S. siamensis* genome and were located on different chromosomes, suggesting that segmental duplications in these locations potentially contributed to expansion of the *SsWRKY* family. Moreover, three collinear maps of *S. siamensis* were constructed with three representative species, and it was shown that 56 *SsWRKY* genes (nearly 95%) were orthologous between *CsWRKY* genes (Appendix A), implying that segmental duplication of *WRKY* genes might have occurred in diploid plants.

A gene’s expression profile can often be related to its function. In this study, 59 *SsWRKYs* were analyzed in four different tissues of *S. siamensis*; root, stem, tuber, and leaf. Among *SsWRKYs*, 22 *SsWRKY* genes were expressed in the roots (Figure 7). These results concur with research reported for other species, such as *A. thaliana* [46], *A. truncatum* [19], and *C. sativus* [26]. It may be that roots, as an important organ for absorbing water and nutrients, respond first to stress when a plant is subjected to drought, high salt, or other stresses. For example, *AtWRKY33* is a negative regulator that mediates Pi-deficiency-induced remodeling of root architecture [47]; *AtWRKY6* and *AtWRKY23* were shown to regulate root growth and development [48]. SsWRKY12, SsWRKY21, and SsWRKY52 are homologous genes of *AtWRKY6*, *AtWRKY23*, and *AtWRKY33*. These genes may play key roles in root growth and development, and may perform a similar function in *S. siamensis* as they do in *A. thaliana. SsWRKY46* is more highly expressed in stems than in other tissues, indicating that some *SsWRKY* genes may be involved in specific organ responses including stress. These results imply that *SsWRKY* genes showing different patterns of expression in different tissues are involved in numerous biological metabolism processes in *S. siamensis*.

The possible regulatory function of SsWRKY protein can be better understood using interaction network analysis of WRKY proteins from model plants. Five SsWRKYs (SsWRKY11, SsWRKY12, SsWRKY38, SsWRKY53, SsWRKY54) with high sequence similarity with AtWRKY62, AtWRKY33, AtWRKY30, AtWRKY40, and AtWRKY46 were identified as potential central nodes of the interaction network. Studies have reported that AtWRKY40 (homolog of SsWRKY53) can modulate the expression of stress-responsive nuclear genes encoding mitochondrial and chloroplast proteins [49]. Also, AtWRKY40 can alter resistance to pathogens. Notably, SsWRKY12 is also a node in the interaction network diagram. AtWRKY33 (homolog of SsWRKY12) has been proposed to interact with AtWRKY30 and AtWRKY15 (homolog of SsWRKY38, 22). Related studies showed that overexpression of AtWRKY33 in Arabidopsis could improve plants’ salt-tolerance [50]. Furthermore, SsWRKY47 was predicted to interact with SsWRKY11, SsWRKY54, AtWRKY70 (homolog of SsWRKY47), and AtWRKY46 (homolog of SsWRKY54), and to play a key role in the regulation of brassinolide in plant growth, development, and drought tolerance [16]. In addition, a strong interaction between SsWRKY3 and SsWRKY39 was predicted. AtWRKY1 (homolog of SsWRKY3) negatively regulates plants’ defense responses to Pst DC3000 through SA signaling pathways [51]. It is known that homologous proteins with similar sequences from different plants may have similar functions. In general, based on previous studies and prediction results of protein-function interaction, it can be inferred that the corresponding *SsWRKY* gene in *S. siamensis* may also have similar functions, and that closely related SsWRKY genes may have close functional relationships.

The WRKY gene family is especially associated with responses to biotic and abiotic stress [52]. Previous studies have shown that WRKY genes function in response to cold stress in many plants. For example, AtWRKY34 negatively effects the CBF-mediated cold response pathway [53]. In *Vitis vinifera*, cold stress induced the most rapid upregulation of VvWRKY genes of all the tested abiotic stress treatments [54]. In *Brassica campestris*, BcWRKY46 is induced by cold stress and ABA, improving the plant’s tolerance of low temperature by activating related genes in the ABA signal pathway [55]. VbWRKY32 positively regulates the expression level of genes in response to cold, which increases antioxidant activity and maintains membrane stability, improving survival ability under cold stress [56]. In this study, 15 *SsWRKY* genes that contained one LTR *cis*-acting element in the promoter region were surveyed for their response to cold-temperature stress in leaf tissue (Figure 8). The results showed that all had altered expression throughout the experiment, and that nine *SsWRKY* genes were maximally expressed under cold stress after 48 h. Similar expression patterns were also found in *V. vinifera* [57]. In addition, several *SsWRKY* genes were highly expressed at 24 h, returning to the initial expression level after 48 h cold treatment, indicating that these *SsWRKY* transcription factors may function variously at different periods of the stress response. Although WRKYs have been observed to function in many plants in response to low temperature, the mechanism of how WRKYs respond to cold signals and regulate the expression of downstream genes remains largely unknown. Further research is required to demonstrate the function of these genes in relation to low temperatures and their involvement in cold signal pathways.

## 4. Materials and Methods

### 4.1. Plant Materials and Stress Treatment

*S. siamensis* plants used in this research were collected from Guangxi Academy of Agricultural Sciences (108.24 E;22.84 N) (Nanning, China). Tissue culture seedlings of *S. siamensis* were placed in a chamber with a mean temperature of 25.0 ± 1.0 °C, relative humidity of 60% ± 10%, and a day/light cycle of 12/12 h. For the cold treatment, *S. siamensis* seedlings were placed in low-temperature refrigerator at 4 °C and samples were gathered at 6, 12, 24, 36, and 48 h with 0 h as control. The leaves of six tissue culture seedlings were collected and mixed to provide one sample. The samples were snap frozen in liquid nitrogen and then stored at −80 °C freezer to extract total RNA.

### 4.2. Sequence Retrieval and Candidate SsWRKYs Identification

The *S. siamensis* genome files (*S._siamensis*.gff, each_chr.stats, and *S._siamensis_*genome.fa) were prepared in order to identify *SsWRKY* genes. The WRKY protein sequences from *A. thaliana* were downloaded from the TAIR database (https://www.arabidopsis.org/index.jsp, accessed on 20 October 2022)as the query sequences for identification of homologous *SsWRKY* sequences, using TBtools software. The HMM file of the WRKY domain (PF03106) was downloaded from the Pfam database (http://pfam.xfam.org/, accessed on 21 October 2022) and candidate *SsWRKY* genes were identified using HMMER3.0 (http://hmmer.janelia.org/, accessed on 21 October 2022) with E-value < 1.0 × 10^−5^ [36]. Then, two online software programs, PfamScan (https://www.ebi.ac.uk/Tools/pfa/pfamscan/, accessed on 21 October 2022) and SMART (http://smart.embl-heidelberg.de/, accessed on 21 October 2022) were employed to identify whether all candidate SsWRKY proteins had complete WRKY DNA binding domains including the conserved WRKYGQK/WRKYGKK and zinc-finger motif [58]. Furthermore, we made use of the online website ExPASy (https://web.expasy.org/protparam/, accessed on 23 October 2022) to calculate the amino acid length, molecular weight (MW), theoretical isoelectric point (pI), and instability index of all candidate SsWRKY proteins, and the online website WoLF PSORT (https://wolfpsort.hgc.jp/, accessed on 23 October 2022) to predict their subcellular localization [59,60].

### 4.3. Multiple Protein Sequence Alignment and Phylogenic Tree Construction

Based on the conserved domain of the *SsWRKY* protein, the family was divided into different groups using DNAMAN7.0 software. The ClusterW tool in MEGA7.0 software was applied to perform multiple protein sequence alignment analysis of *WRKYs* from *S. siamensis* and *A. thaliana*, using default parameter values and the neighbor-joining method to construct a phylogenetic evolution tree, with 1000 bootstrap iterations [61]. Subsequently, the graphic was transformed using Illustrator 2020 (v24.0.0.330). Based on the reported classification of *WRKYs* from *A. thaliana*, the *SsWRKYs* were classified into relevant groups.

### 4.4. Gene Structure and WRKY Protein Conserved Motif Analysis

The DNA and CDS sequences of the *SsWRKY* gene family were screened from the whole genome sequence and gene annotation file of *S. siamensis*. Then, the exon–intron structures of *SsWRKYs* were analyzed and visualized using TBtools software. The MEME v5.1.1 online program (https://meme-suite.org/meme/, accessed on 23 October 2022) was used to predict the conserved motifs of WRKY protein, with the maximum number of motifs set to 10 [62]. The diagrammatic sketches of gene structures and conserved motifs were re-edited using Photoshop 2022 software.

### 4.5. Chromosomal Location, Gene Duplication and Collinearity Analysis

All *SsWRKY* genes were mapped to the 14 chromosomes of *S. siamensis* according to their positions on the *S. siamensis* genome sequence, which was displayed using Circos (http://circos.ca/, accessed on 23 October 2022). Blastp was used to analyze the gene duplication events with default parameters and MCSanX to detect collinearity relationship between *S. siamensis* and three other species (*Oryza sativa*, *A. thaliana*, *C. sativus*) [63,64].

### 4.6. Promoter Cis-Acting Element Analysis of SsWRKY Gene Family

TBtools software (GTF/GFF3 sequences extractor) was applied to extract the 2000 bp sequence upstream of the CDS sequences of *S. siamensis*, and the PlantCARE database (http://bioinformatics.psb.ugent.be/webtools/plantcare/html/, accessed on 23 October 2022) was used for prediction of cis-acting elements in the promoter regions [65].

### 4.7. Protein Interaction Network and Gene Co-Expression of SsWRKYs

For the protein interaction network, String (http://string-db.org/, accessed on 23 October 2022) was employed to analyze the SsWRKY protein interaction with reference to AtWRKY protein, with the parameter threshold set to 0.15 [66]. *A. thaliana* was selected as the model plant to analyze the co-expression of *SsWRKY*. The network was visualized using Cytoscape version 3.7.0 [67].

### 4.8. Expression Analysis of SsWRKY in Different Tissues

In order to analyze the expression patterns of *SsWRKY* genes in different tissues, the RNA-seq data of *SsWRKY* genes were gathered from the transcriptome database (unpublished), which includes gene expression levels in roots, tubers, leaves, and stems. The values for fragments per kilobase of transcript per million fragments mapped (FPKM) were utilized to calculate the genome-wide expression of the *SsWRKY* genes, which was then visualized on a heatmap using TBtools software, with the expression levels shown by different colors from blue to red. 

### 4.9. Total RNA Extraction, cDNA Synthesis, and qRT-PCR Validation

Samples consisting of 50–100 mg of tissue in liquid nitrogen were fully milled to a fine powder, and total RNA was extracted using the Trizol method as previously reported [68]. The purity and concentration of RNA was measured using a Micro Drop system (BIO-DL, Shanghai, China), and the RNA integrity was checked on 1% agarose gels. Total RNA (1000 ng) was used for reverse transcription using an Evo M-MLV RT mix kit for qPCR, with gDNA Clean (Accurate Biology, Changsha, China) to remove genomic DNA contamination, at 20 μL volume according to the manufacturer’s instructions. The cDNA samples were diluted 5-fold with RNase-free water to use as templates for qRT-PCR analysis. Specific primer pairs were designed using Beacon Designer 7.0 software and were restricted to primer sequences of 18–25 bp, amplicon length 80–150 bp, melting temperature (Tm) 55–60 °C, and GC content of 45–55%; the primers used for this study are shown in Appendix A. The reaction was carried out in 96-well plates with a DiceTM Real Time System III thermal cycler (Takara, Japan) and a 20 μL reaction system: 2 × SYBR Green Pro Taq HS premix 10 μL (Vazyme, China), 2 μL cDNA, 0.4 μL each of forward primer and reverse primer, 7.2 μL RNase free water. The amplification reaction procedure was: 95 °C, 30 s; 95 °C, 5 s; 60 °C, 30 s, for 40 cycles. Each qRT-PCR analysis was performed with three technical replicates and three biological replications, the relative expression level was calculated using the 2^−∆∆CT^ method. The relative expression under cold-stress treatment of *SsWRKYs* containing LTR-motifs was visualized using GraphPad Prism 9 software.

## 5. Conclusions

We identified 59 *WRKY* genes distributed on 14 chromosomes in *S. siamensis* based on a genome-wide analysis. These 59 *SsWRKY* genes were divided into three major groups based on their phylogenetic relationships. The comparison of *WRKY* genes in *S. siamensis* with three diverse species helped to visualize the expansion and contraction of the *SsWRKY* gene family through its evolution. Expression profiles based on the RNA-seq data from four different tissues showed that these genes have complex expression profiles and may therefore have diverse but important roles in roots, stems, tubers, and leaves. Furthermore, a protein–protein interaction network was predicted for 18 of 59 SsWRKYs. Finally, 15 *SsWRKY* genes with LTR *cis*-acting elements revealed altered and complex expression in response to cold stress. Our study describes a functional framework for WRKY genes, which can provide the basis for further exploration of the function and regulatory mechanism of *WRKY* genes in *S. siamensis*.

## Figures and Tables

**Figure 1 plants-12-00288-f001:**
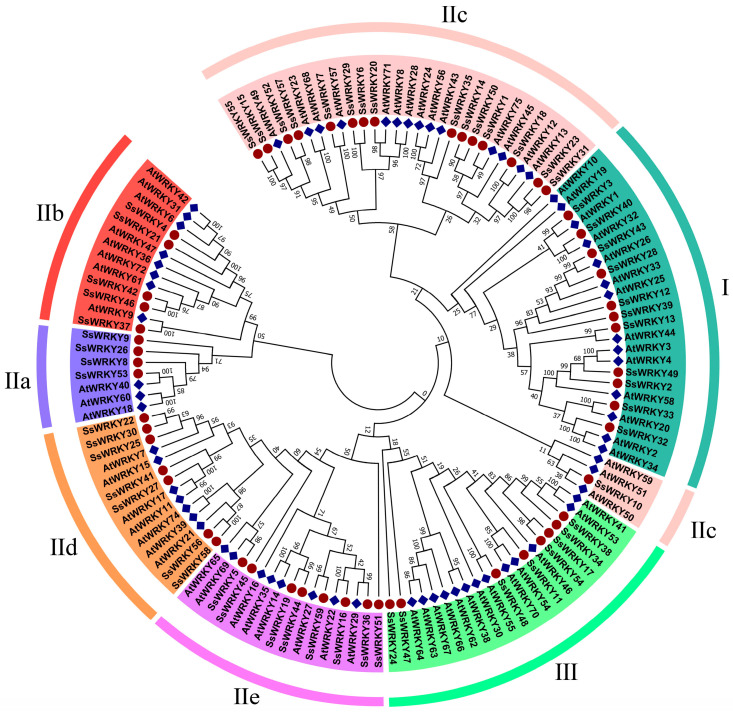
Phylogenetic tree of WRKY proteins from *A. thaliana* and *S. siamensis*. *S. siamensis* and 70 *A. thaliana* WRKY proteins were aligned by ClustW, and the phylogenetic tree was constructed using the neighbor-joining method and MEGA 7.0 with 1000 bootstrap replicates. The WRKY proteins from different groups are indicated with different colors, proteins from *A. thaliana* and *S. siamensis* are denoted by red circles and blue squares.

**Figure 2 plants-12-00288-f002:**
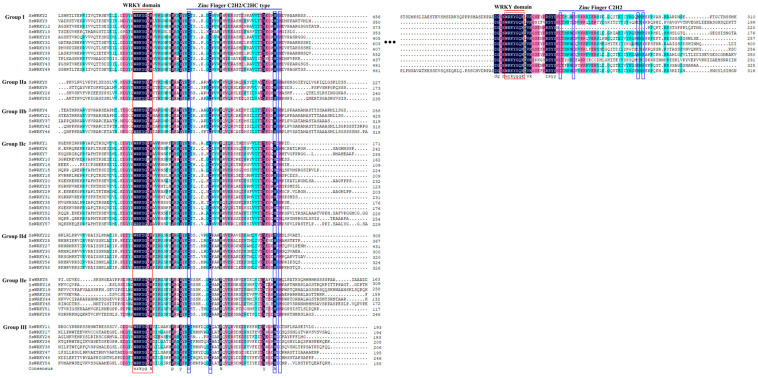
Multiple sequence alignments of the WRKY domain amino acid sequences of 59 SsWRKY proteins, visualized using the DNAMAN7.0 program. The highly conserved regions in each groups are shown in dark blue. The red lines represent the WRKY domains, and the blue lines indicate the conserved zinc finger motifs.

**Figure 3 plants-12-00288-f003:**
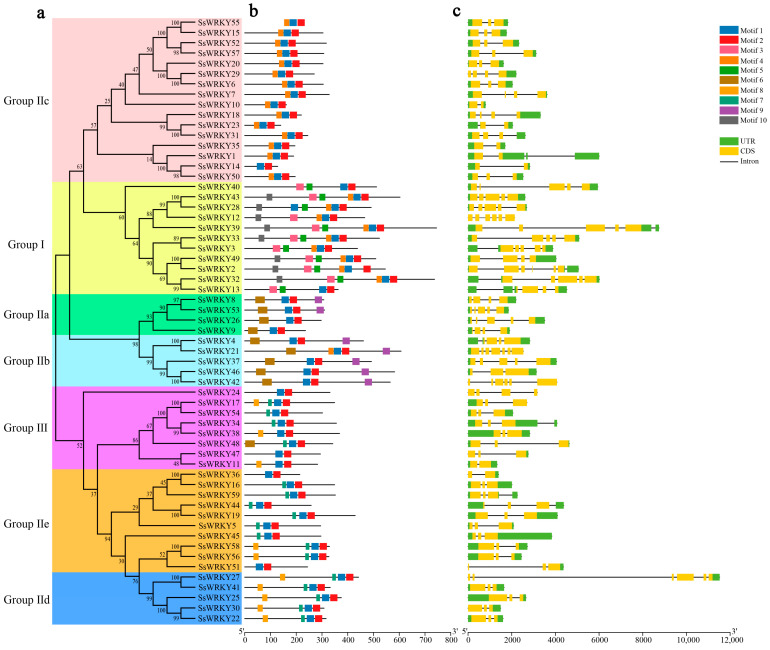
Gene structure and conserved motif analysis of *SsWRKY* gene family. (**a**) Phylogenetic tree of SsWRKYs constructed using MEGA7.0 software; (**b**) Motif compositions of SsWRKY proteins. Ten motifs are represented by different colored boxes numbered 1–10; (**c**) Exon–intron structural analysis of *SsWRKY* genes. Green boxes, yellow boxes, and black lines represent UTR, CDS, and introns, respectively; (**d**) Sequence logos foor motif 1-10.

**Figure 4 plants-12-00288-f004:**
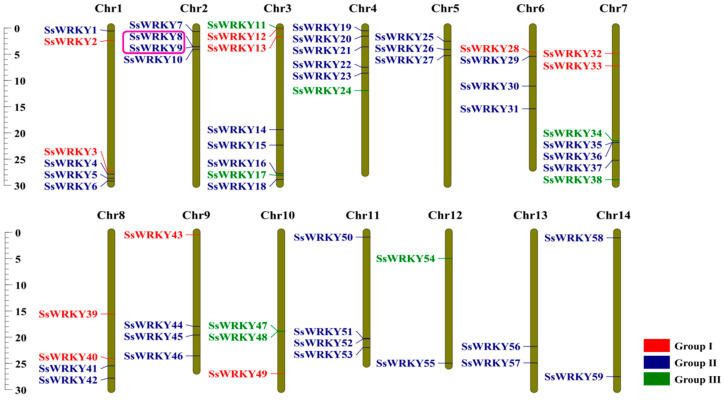
*SsWRKY* gene distribution across 14 chromosomes of *S. siamensis*. Tandem duplicated genes are marked with a pink box. Vertical bars represent chromosomes. The different groups of *SsWRKY* genes are colored red, blue, or green. Base-pair positions are indicated on the left.

**Figure 5 plants-12-00288-f005:**
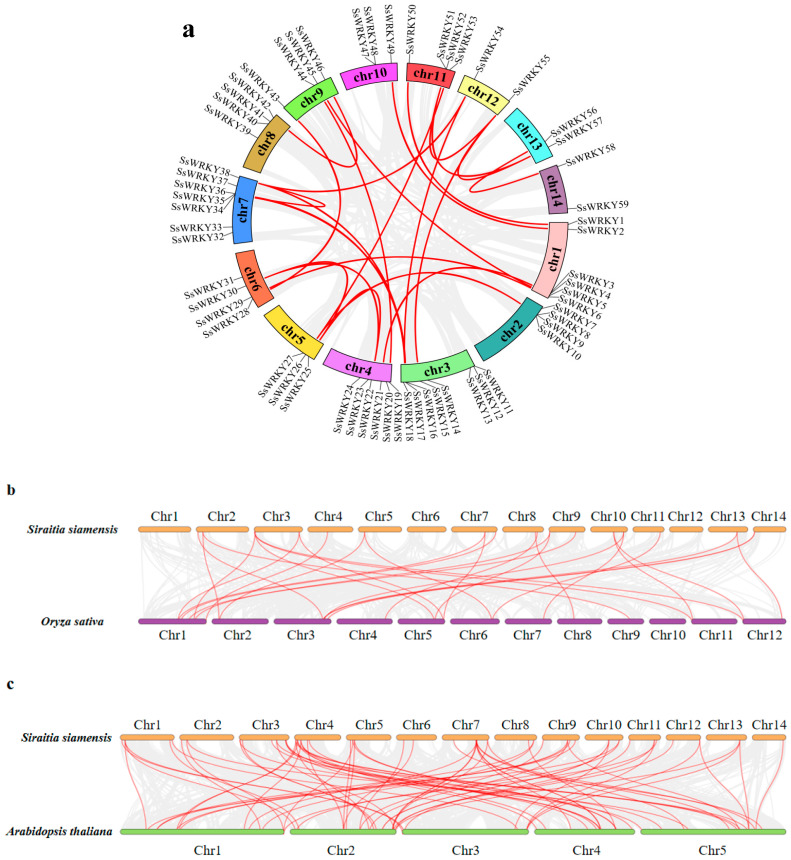
Chromosome-level analysis of the *SsWRKY* gene family in the *S. siamensis* genome assembly. (**a**) Chromosomal location of *SsWRKY* genes and their collinear relationships are shown in the circle diagram. Gray lines indicate all synteny blocks in the *S. siamensis* genome, and red lines represent segmentally duplicated gene pairs. Chromosomes 1–9 are denoted by different colors; (**b**–**d**) Collinearity analysis of *WRKY* genes between *S. siamensis* and *O. sativa*, *A. thaliana*, and *C. sativus*. The syntenic gene pairs of *SsWRKY* genes are tagged with red lines, and the collinear blocks are presented by gray lines.

**Figure 6 plants-12-00288-f006:**
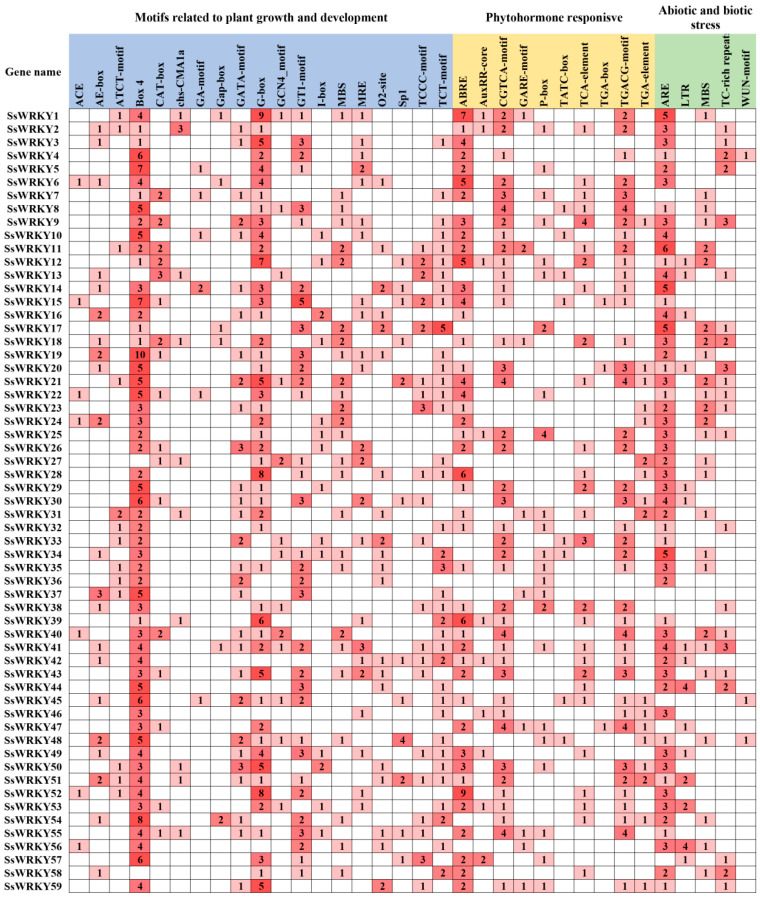
*Cis*-acting elements analysis in the promoter region of *SsWRKY* genes. The number of *cis*-acting elements are shown by the digits. The greater the quantity, the darker the color. Light blue, yellow, and green represent three categories of *cis*-acting elements related to plant growth and development, phytohormone response, and abiotic and biotic stress.

**Figure 7 plants-12-00288-f007:**
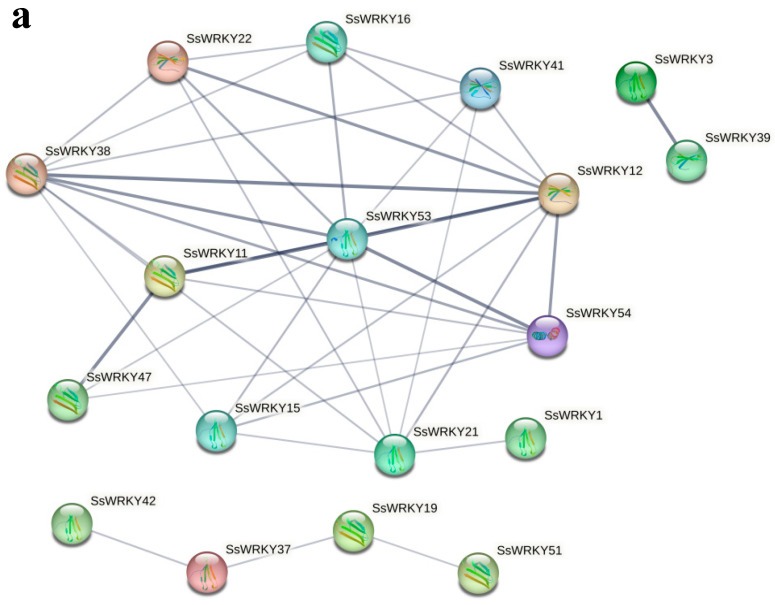
Functional interaction network diagram of SsWRKY protein and co-expression of *SsWRKY* genes. (**a**) Interaction network diagram of SsWRKY proteins. The thicker the gray connecting line, the stronger is the predicted interaction between the two proteins; (**b**) co-expression analysis of *SsWRKY* genes. * *p* < 0.05.

**Figure 8 plants-12-00288-f008:**
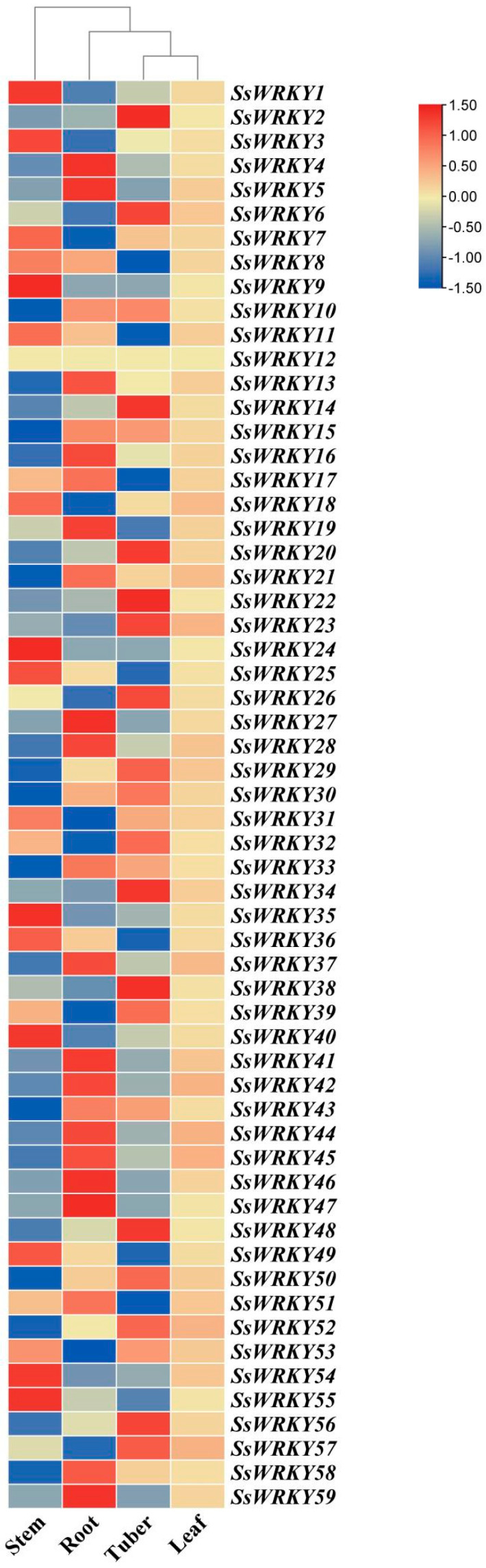
Expression patterns of *SsWRKY* genes in four different tissues of *S. siamensis*. RPKM value was assessed to investigate the expression levels of *SsWRKY* genes in stems, roots, tubers, and leaves. The color variation from blue to red represents low to high expression. Our findings will provide references for the function and regulation mechanism of the *WRKY* gene family in other plants.

**Figure 9 plants-12-00288-f009:**
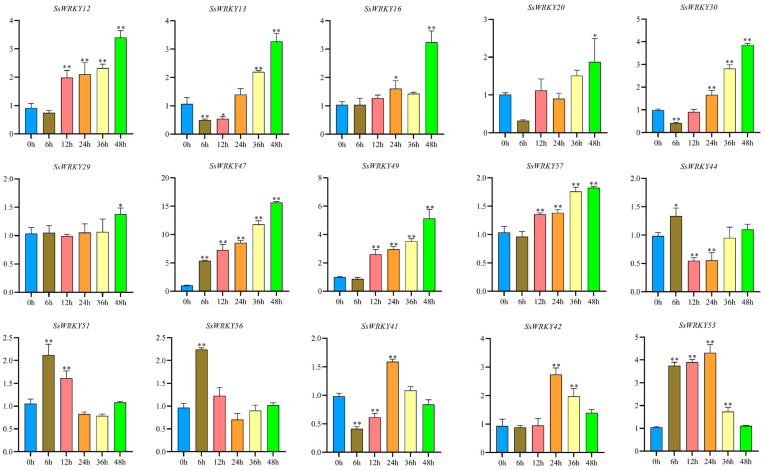
qRT-PCR analysis of *SsWRKY* genes under cold stress. The Y-axes represent the relative expression and X-axes represent the duration of the cold treatment. Data represents the mean ± SD of three technical repetitions. * *p* < 0.05, ** *p* < 0.01.

**Table 1 plants-12-00288-t001:** Information for the WRKY transcription factor family in *S. siamensis*.

Gene Name	Gene ID	Gene Start	Gene End	Protein Length (aa)	Molecular Weight	Theoretical pI	Instability Index	Subcellular Localization
*SsWRKY1*	SsiaChr1G00020800.1	751919	757923	190	21,462.72	5.92	45.09	nucleus
*SsWRKY2*	SsiaChr1G00023090.1	2499410	2504472	546	60,090.33	7.67	58.86	nucleus
*SsWRKY3*	SsiaChr1G00040860.1	26599342	26603231	438	47,993.39	6.81	38.10	nucleus
*SsWRKY4*	SsiaChr1G00041580.1	27084596	27087427	461	49,397.04	8.40	45.55	nucleus
*SsWRKY5*	SsiaChr1G00042810.1	27853616	27855704	295	31,790.99	5.50	59.35	nucleus
*SsWRKY6*	SsiaChr1G00043600.1	28351114	28353153	305	34,321.93	7.10	71.15	nucleus
*SsWRKY7*	SsiaChr2G00001280.1	860173	863790	328	36,013.85	6.21	59.23	nucleus
*SsWRKY8*	SsiaChr2G00004890.1	3652228	3654428	307	33,802.91	8.57	45.82	nucleus
*SsWRKY9*	SsiaChr2G00004900.1	3661109	3663016	236	26,486.98	9.57	70.01	nucleus
*SsWRKY10*	SsiaChr2G00005450.1	4137409	4138214	162	18,458.21	5.17	39.23	nucleus
*SsWRKY11*	SsiaChr3G00044220.1	197881	199214	283	31,834.41	5.65	58.91	nucleus
*SsWRKY12*	SsiaChr3G00044550.1	429901	432039	466	50,804.16	6.15	61.60	nucleus
*SsWRKY13*	SsiaChr3G00046440.1	1628752	1633276	363	40,551.22	9.38	36.13	nucleus
*SsWRKY14*	SsiaChr3G00054430.1	17366784	17369617	128	14,590.29	6.90	57.03	nucleus
*SsWRKY15*	SsiaChr3G00056180.1	19990778	19992548	304	33,509.51	6.61	66.41	nucleus
*SsWRKY16*	SsiaChr3G00062240.1	24823802	24825812	349	37,785.78	5.70	58.09	nucleus
*SsWRKY17*	SsiaChr3G00062690.1	25101188	25103885	349	38,684.93	4.88	54.01	nucleus
*SsWRKY18*	SsiaChr3G00063680.1	25790770	25794091	220	24,863.42	8.74	45.10	nucleus
*SsWRKY19*	SsiaChr4G00082080.1	664352	668450	429	46,841.51	5.15	52.61	nucleus
*SsWRKY20*	SsiaChr4G00083060.1	1571966	1573595	304	33,330.52	5.93	52.89	nucleus
*SsWRKY21*	SsiaChr4G00085620.1	3371552	3374085	607	65,921.96	6.77	46.81	nucleus
*SsWRKY22*	SsiaChr4G00090590.1	6827223	6828826	316	34,224.58	9.59	37.56	nucleus
*SsWRKY23*	SsiaChr4G00091770.1	7817291	7819338	140	16,308.49	9.92	38.24	nucleus
*SsWRKY24*	SsiaChr4G00095300.1	10761531	10764705	331	36,407.8	9.15	67.77	nucleus
*SsWRKY25*	SsiaChr5G00105720.1	2172446	2175101	375	40,443.59	9.54	47.46	nucleus
*SsWRKY26*	SsiaChr5G00107540.1	3423281	3426794	297	34,150.07	9.39	51.45	nucleus
*SsWRKY27*	SsiaChr5G00108780.1	4344810	4356332	442	47,253.55	9.06	52.87	nucleus
*SsWRKY28*	SsiaChr6G00152850.1	3845799	3848486	491	54,198.42	9.11	61.97	nucleus
*SsWRKY29*	SsiaChr6G00153650.1	4474657	4476865	270	30,054.3	6.83	57.17	nucleus
*SsWRKY30*	SsiaChr6G00159600.1	8993505	8995000	308	33,360.81	9.38	51.73	nucleus
*SsWRKY31*	SsiaChr6G00161490.1	12439816	12442437	245	27,878.37	9.08	57.29	nucleus
*SsWRKY32*	SsiaChr7G00067510.1	4245861	4251877	737	79,659.14	5.94	55.04	nucleus
*SsWRKY33*	SsiaChr7G00068370.1	6214402	6219491	523	57,736.24	6.89	45.11	nucleus
*SsWRKY34*	SsiaChr7G00073090.1	18164282	18168358	356	40,012.95	5.73	59.24	nucleus
*SsWRKY35*	SsiaChr7G00073400.1	18453322	18455025	195	21,397.1	7.00	44.92	nucleus
*SsWRKY36*	SsiaChr7G00073410.1	18456961	18458359	214	24,010.52	4.95	58.25	nucleus
*SsWRKY37*	SsiaChr7G00076520.1	21283722	21287776	492	55,112.31	5.61	51.35	nucleus
*SsWRKY38*	SsiaChr7G00080620.1	24382740	24385570	368	40,365.31	5.82	50.80	nucleus
*SsWRKY39*	SsiaChr8G00123950.1	11741204	11749949	745	81,911.12	5.51	48.40	nucleus
*SsWRKY40*	SsiaChr8G00130410.1	18147025	18152970	512	56,211.49	6.28	57.55	nucleus
*SsWRKY41*	SsiaChr8G00131810.1	19170655	19172307	332	36,140.06	9.55	52.72	nucleus
*SsWRKY42*	SsiaChr8G00134260.1	20940110	20944180	565	60,663.57	6.89	59.63	nucleus
*SsWRKY43*	SsiaChr9G00178500.1	390249	392867	603	65,652.99	7.28	62.23	nucleus
*SsWRKY44*	SsiaChr9G00184810.1	13512314	13516697	258	28,929.78	6.15	44.44	nucleus
*SsWRKY45*	SsiaChr9G00186390.1	14756723	14760562	296	33,177.89	9.44	64.08	nucleus
*SsWRKY46*	SsiaChr9G00190810.1	17742607	17745742	582	62,261.62	6.87	41.80	nucleus
*SsWRKY47*	SsiaChr10G00168450.1	12831606	12834367	294	32,742.6	6.01	52.60	nucleus
*SsWRKY48*	SsiaChr10G00168460.1	12838628	12843264	342	37,504.25	7.07	51.72	nucleus
*SsWRKY49*	SsiaChr10G00175160.1	18327226	18331262	509	55,833.51	8.78	55.31	nucleus
*SsWRKY50*	SsiaChr11G00215290.1	661155	663681	196	21,975.74	9.28	31.80	nucleus
*SsWRKY51*	SsiaChr11G00221880.1	13766286	13770658	244	27,386.47	4.83	55.79	nucleus
*SsWRKY52*	SsiaChr11G00221950.1	13823287	13825620	317	34,872.95	6.06	54.27	nucleus
*SsWRKY53*	SsiaChr11G00223210.1	14943982	14945840	310	34,106.25	8.20	47.54	nucleus
*SsWRKY54*	SsiaChr12G00195640.1	3637484	3639541	302	33,579.64	5.30	50.52	nucleus
*SsWRKY55*	SsiaChr12G00203270.1	18044155	18045982	338	37,423.83	6.18	64.65	nucleus
*SsWRKY56*	SsiaChr13G00140450.1	15745104	15747555	327	36,421.34	9.65	62.67	nucleus
*SsWRKY57*	SsiaChr13G00142910.1	18011118	18014237	307	34,072.08	7.05	62.00	nucleus
*SsWRKY58*	SsiaChr14G00204380.1	659304	662023	329	36,799.93	9.69	47.81	nucleus
*SsWRKY59*	SsiaChr14G00213550.1	16567231	16569495	352	38,011.63	5.78	60.17	nucleus

## Data Availability

All WRKY protein sequences are provided in Appendix A.

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
