# Peer review of "Genome-Wide Identification and Expression Analysis of WRKY Transcription Factors in Siraitia siamensis"

_plants, 2023, doi:10.3390/plants12020288_

Round 1

Reviewer 1 Report

The manuscript "Genome-wide Identification and Expression Analysis of WRKY Transcription Factors in Siraitia siamensis" describes the WRKY gene family from S. siamensis. RT-qPCR was used to identify 15 SsWRKYs with LTR motifs under cold stress. This manuscript provides a foundation for further study on the function and regulatory mechanism of SsWRKY gene family in S. siamensis. Hence, the present study is useful, clear, and solid. However, a few improvements should be modified for acceptance as follows.

1.    The Latin names of species should be italicized. Please check thoroughly this manuscript.

2.    In Table 1, the figure in Theoretical pI and Instability index should be retained at two decimal places.

3.    In Figure 1, the SsWRKY31 should be classified into group IIc but not the group I.

4.    The title of the 2.2Classification and Phylogenetic Relationships of SsWRKY Genes” should be changed since this part was related to the analysis of the SsWRKY protein sequences.

5.    In lines 288 and 300, Figure 7 should be changed to Figure 8, and Figure 8 should be Figure 9.

6.    In Figure 9, the significance analysis of supplementary differences among different treatments should be supplemented. The 15 SsWRKY genes should be analyzed but not UrWRKY.

Reviewer 2 Report

1.The table is recommended to be placed in the attachment.

2. The difference test is missing in Figure 9.

Reviewer 3 Report

The authors perform a comprehensive bioinformatics study of WRKY Transcription Factors in Siraitia siamensis and use qPCR to determine their role under environmental stimuli. The manuscript is in fairly; however, I believe it does not match the quality of the journal. Therefore I recommend conducting further experiments, such as silencing/overexpression to determine the precise functioning of genes under various conditions, and yeast assays (Y1H, Y2H, EMSA) to perform interaction assays. I recommend rejecting the article and reconsidering it for the reviewing process after the addition of further assays.

Round 2

Reviewer 3 Report

Without further validation, papers doesn't have much novel findings. in its present form, it can not be accepted keeping in view the potential of journal standard.

Author Response

Please see the attachment. Thank you for your suggestions.